# Learning Hash Codes via Hamming Distance Targets

## Abstract

We present a powerful new loss function and training scheme for learning binary hash codes with any differentiable model and similarity function. Our loss function improves over prior methods by using log likelihood loss on top of an accurate approximation for the probability that two inputs fall within a Hamming distance target. Our novel training scheme obtains a good estimate of the true gradient by better sampling inputs and evaluating loss terms between all pairs of inputs in each minibatch. To fully leverage the resulting hashes, we use multi-indexing. We demonstrate that these techniques provide large improvements to a similarity search tasks. We report the best results to date on competitive information retrieval tasks for ImageNet and SIFT 1M, improving MAP from 73% to 85% and reducing query cost by a factor of 2-8, respectively.

## 1 Introduction

Many information retrieval tasks rely on searching high-dimensional datasets for results similar to a query. Recent research has flourished on these topics due to enormous growth in data volume and industry applications Wang et al. (2016). These problems are typically solved in either two steps by computing an embedding and then doing lookup in the embedding space, or in one step by learning a hash function. We call these three problems the data-to-embedding problem, the embedding-to-results problem, and the data-to-results problem. There exists an array of solutions for each one.

Models that solve data-to-embedding problems aim to embed the input data in a space where proximity corresponds to similarity. The most commonly chosen embedding space is $\mathbb{R}^n$, in order to leverage lookup methods that assume Euclidean distance. Recent methods employ neural network architectures for embeddings in specific domains, such as facial recognition and sentiment analysis Schroff et al. (2015); Mikolov et al. (2013).

Once the data-to-embedding problem is solved, numerous embedding-to-results strategies exist for similarity search in a metric space. For this step, the main challenge is achieving high recall with low query cost. Exact $k$-nearest neighbors (KNN) algorithms achieve 100% recall, finding the $k$ closest items to the query in the dataset, but they can be prohibitively slow. Brute force algorithms that compare distance to every other element of the dataset are often the most viable KNN methods, even with large datasets. Recent research has enabled exact KNN on surprisingly large datasets with low latency Johnson et al. (2017). However, the compute resources required are still large. Alternatives exist that can reduce query costs in some cases, but increase insertion time. For instance, $k$-d trees require $O(\log N)$ search time on average with a high constant, but also require $O(\log N)$ insertion time on average.

Approximate nearest neighbors algorithms solve the embedding-to-results problem by finding results that are likely, but not guaranteed to be among the $k$ closest. Similarly, approximate near-neighbor algorithms aim to find most of the results that fall within a specific distance of the query's embedding. These tasks (ANN) are generally achieved by hashing the query embedding, then looking up and comparing results under hashes close to that hash. Approximate methods can be highly advantageous by providing orders of magnitude faster queries with constant insertion time. Locality-sensitive hashing (LSH) is one such method that works by generating multiple, randomly-chosen hash functions for each input. Each element of the dataset is inserted into multiple hash tables, one for each hash function. Queries can then be made by checking all hash tables for similar results. Another approach is quantization, which solves ANN problems by partitioning the space of inputs into

buckets. Each element of the dataset is inserted into its bucket, and queries are made by selecting from multiple buckets close to the query.

Data-to-results methods determine similarity between inputs and provide an efficient lookup mechanism in one step. These methods directly compute a hash for each input, showing promise of simplicity and efficiency. Additionally, machine learning methods in this category train end-to-end, by which they can reduce inefficiencies in the embedding step. There has been a great deal of recent research into these methods in topics such as content-based image retrieval (CBIR). In other topics such as automated scene matching, hand-chosen hash functions are common Ansari & Mohammed (2015). But despite recent focus, data-to-results methods have had mixed results in comparison to data-to-embedding methods paired with embedding-to-results lookup Wang et al. (2018); Klein & Wolf (2017).

We assert the main reason data-to-results methods have sometimes underperformed is that training methods have not adequately expressed the model's loss. Our proposed approach trains neural networks to produce binary hash codes for fast retrieval of results within a Hamming distance target. These hash codes can be efficiently queried within the same Hamming distance by multi-indexing Norouzi et al. (2012).

## 1.1 RELATED WORK

Additional context in quantization and learning to hash is important to our work. Quantization is considered state-of-the-art in ANN tasks Wang et al. (2018). There are many quantization approaches, but three are particularly noteworthy: iterative quantization (ITQ) Gong & Lazebnik (2013), product quantization (PQ) Jégou et al. (2011), and multi-scale quantization (MSQ) Wu et al. (2017). Iterative quantization learns to produce binary hashes by first reducing dimensionality and then minimizing a *quantization loss* term, a measure of the amount of information lost by quantizing. ITQ uses principal component analysis for dimensionality reduction and $||\text{sgn}(v) - v||_2$ for a quantization loss term, where $v$ is the pre-binarized output and $\text{sgn}(v)$ is the quantized hash. It then minimizes quantization loss by alternately updating an offset and then a rotation matrix for the embedding. PQ is a generally more powerful quantization method that splits the embedding space $\mathbb{R}^n$ into $\mathbb{R}^{n/M} \times \mathbb{R}^{n/M} \times \ldots \mathbb{R}^{n/M}$. A $k$-means algorithm is run on the embedding constrained to each $\mathbb{R}^{n/M}$ subspace, giving $k$ Voronoi cells in each subspace for a total of $k^m$ hash buckets. MSQ builds on PQ by separately quantizing the magnitude and directions of each vector, breaking $\mathbb{R}^n$ into $\mathbb{R} \times S^{n-1}$.

Recent methods that learn to hash end-to-end draw from a few families of loss terms to train binary codes Wang et al. (2018). These include terms for supervised softmax cross entropy between codes Jain et al. (2017), supervised Euclidean distance between codes Liu et al. (2016), and quantization loss terms Zhou et al. (2017). Softmax cross entropy and Euclidean distance losses assume that Hamming distance corresponds to Euclidean distance in the pre-binarized outputs. Some papers try to enforce that assumption in a few different ways. For instance, quantization loss terms aim to make that assumption more true by penalizing networks for producing outputs far from $\pm 1$. Alternative methods to force outputs close to $\pm 1$ exist, such as HashNet, which gradually sharpens sigmoid functions on the pre-binarized outputs. Another family of methods first learns a target hash code for each class, then minimizes distance between each embedding and its target hash code Xia et al. (2014); Lu et al. (2017).

We observed four main shortcomings of existing methods that learn to hash end-to-end. First, cross entropy and Euclidean distance between pre-binarized outputs does not correspond to Hamming distance under almost any circumstances. Second, quantization loss and learning by continuation cause gradients to shrink during training, dissuading the model from changing the sign of any output. Third, methods using target hash codes are limited to classification tasks, and have no obvious extension to applications with non-transitive similarity. Finally, various multi-step training methods, including target hash codes, forfeit the benefit of training end-to-end.

## 1.2 MULTI-INDEXING

Multi-indexing enables search within a Hamming radius $r$ by splitting an $n$-bit binary hash into $m$ substrings of length $n/m$ Norouzi et al. (2012). Technically, it is possible to use any $m \in$

$\{1, \ldots r + 1\}$, but in most practical scenarios the best choice is $m = r + 1$. We consider only this case[1]. Each of these $r + 1$ substrings is inserted into its own reverse index, pointing back to the content and full hash (Algorithm 1). Insertion runtime is therefore proportional to $r + 1$, the number of multi-indices.

Lookup is performed by taking the union of all results for each substring, then filtering down to results within the Hamming radius $r$ (Algorithm 2). This enables lookup within a Hamming radius of $r$ by querying each substring in its corresponding index. Any result within $r$ will match on at least one of the $r + 1$ substrings by pigeonhole principle.

---

**Algorithm 1** Insertion in a multi-index system

---

**Input:** binary hash $h$ and corresponding data $D$
Split $h$ into substrings $h_1, \ldots h_{r+1}$
**for** $i = 1$ **to** $r + 1$ **do**
    Add row with key $h_i$ and data $(h, D)$ to the $i$th index
**end for**

---

**Algorithm 2** Lookup in a multi-index system

---

**Input:** binary hash $h$
Split $h$ into substrings $h_1, \ldots h_{r+1}$
Initialize empty set $S_D$
**for** $i = 1$ **to** $r + 1$ **do**
    Add exact matches for $h_i$ in the $i$th index to $S_D$
**end for**
Filter results with Hamming distance greater than $r$ out of $S_D$
Return $S_D$

---

With a well-distributed hash function, the average runtime of a lookup is proportional to the number of queries times the number of rows returned per query. Norouzi et al. treat the time to compare Hamming distance between codes as constant[2], giving us a query cost of

$$\text{cost} \sim (r + 1) \frac{N}{2^{n/(r+1)}}$$

where $N$ is the total number of $n$-bit hashes in the database. Like Norouzi et al., we recommend choosing $r$ such that $n/(r + 1) \approx \log_2 N$, providing a runtime of

$$\text{cost} \sim \frac{n}{\log_2 N}$$

Space cost to store the dataset is $Nn(r + 1)$, since each substring must point back to its full hash. However, since $n$ is only a very small bit length, this is quite manageable.

We build on this technique in 2.3.

## 2    METHOD

We propose a method of Hamming distance targets (HDT) that can be used to train any differentiable, black box model to hash. We will focus on its application to deep convolutional neural nets trained using stochastic gradient descent. Our loss function's foundation is a statistical model relating pairs of embeddings to Hamming distances.

---

[1]In scenarios with a combination of extremely large datasets, short hash codes, and large $r$, it is more efficient to use $m < r + 1$ substrings and make up for the missing Hamming radius with brute-force searches around each substring. However, since we are learning to hash, it makes more sense to simply choose a longer hash.

[2]A binary code can be treated as a long for $n \leq 64$, giving constant time to XOR bits with another code on x64 architectures. Summing the bits is $O(n)$, but small compared to the practical cost of retrieving a result.

## 2.1 LOSS FUNCTION

### 2.1.1 MOTIVATION

Let $\boldsymbol{y}(\boldsymbol{x}) = (y_1(\boldsymbol{x}), \ldots y_n(\boldsymbol{x}))$ be the model's embedding for an input $\boldsymbol{x}$, and let $\mathcal{X}$ be the distribution of inputs to consider. We motivate our loss function with the following assumptions:

- If $\boldsymbol{x} \sim \mathcal{X}$ is a random input, then $y_i(x) \sim \mathcal{N}(0, 1)$. We partially enforce this assumption via batch normalization of $y_i$ with mean 0 and variance 1.

- $y_i$ is independent of other $y_j$.

Let $\boldsymbol{z}(\boldsymbol{x}) = \boldsymbol{y}(\boldsymbol{x})/||\boldsymbol{y}(\boldsymbol{x})||_2$ be the $L^2$-normalized output vector. Since $\boldsymbol{y}(\boldsymbol{x})$ is a vector of $n$ independent random normal variables, $\boldsymbol{z}(\boldsymbol{x})$ is a random variable distributed uniformly on the hypersphere.

This $L^2$-normalization is the same as SphereNorm Liu et al. (2017) and similar to Riemannian Batch Normalization Cho & Lee (2017). Liu et al. posed the question of why this technique works better in conjunction with batch norm than either approach alone, and our work bridges that gap. An $L^2$-normalized vector of IID random normal variables forms a uniform distribution on a hypersphere, whereas most other distributions would not. An uneven distribution would limit the regions on the hypersphere where learning can happen and leave room for internal covariate shift toward different, unknown regions of the hypersphere.

To avoid the assumption that Euclidean distance translates to Hamming distance, we further study the distribution of Hamming distance given these $L^2$-normalized vectors. We craft a good approximation for the probability that two bits match, given two uniformly random points $\boldsymbol{z}^i, \boldsymbol{z}^j$ on the hypersphere, conditioned on the angle $\theta$ between them.

We know that $\boldsymbol{z}^i \cdot \boldsymbol{z}^j = \cos(\theta)$, so the arc length of the path on the unit hypersphere between them is $\arccos(\boldsymbol{z}^i \cdot \boldsymbol{z}^j)$. A half loop around the unit hypersphere would cross each of the $n$ axis hyperplanes (i.e. $z_k = 0$) once, so a randomly positioned arc of length $\theta$ crosses $n\theta/\pi$ axis hyperplanes on average (Figure 1). Each axis hyperplane crossed corresponds to a bit flipped, so the probability that a random bit differs between these vectors is

$$P_{ij} = \frac{\arccos\left(\boldsymbol{z}^i \cdot \boldsymbol{z}^j\right)}{\pi}$$

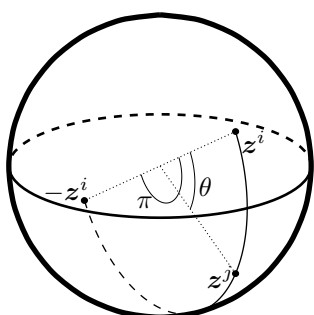

Figure 1: An arc of length $\theta$ on the unit hypersphere starting from a random point in a random direction has probability $\theta/\pi$ for the sign of a particular component to change along its course. In the 3D example above, crossing the great circle implies that the sign of one component differs between $\boldsymbol{z}^i$ and $\boldsymbol{z}^j$.

Given this exact probability, we estimate the distribution of Hamming distance between $\mathrm{sgn}(\boldsymbol{y}^i)$ and $\mathrm{sgn}(\boldsymbol{y}^j)$ by making the approximation that each bit position between the two vectors differs independently from the others with probability $\boldsymbol{P}_{ij}$. Therefore, the probability of Hamming distance being within $r$ is approximately $F(r; n, \boldsymbol{P}_{ij})$ where $F$ is the binomial CDF. This approximation proves to be very close for large $n$ (Figure 2).

Prior hashing research has made inroads with a similar observation, but applied it in the limited context of choosing vectors to project an embedding onto for binarization Ji et al. (2012). Prior quantization research has used the geometry of the hypersphere before, but to choose a projection that minimizes quantization loss Gong et al. (2012). Instead, we apply this idea directly in network training.

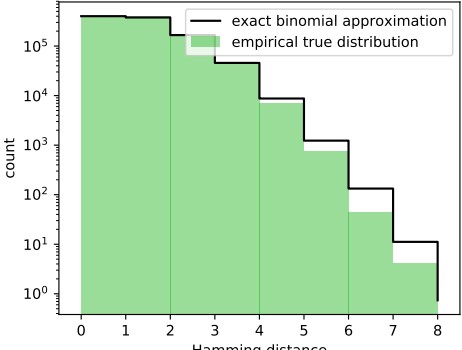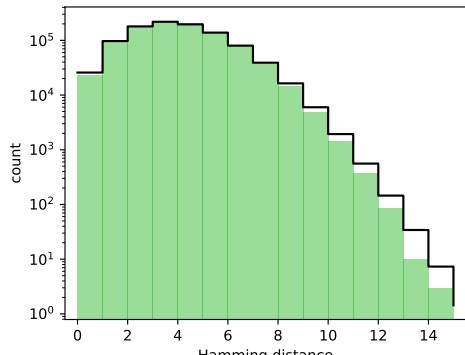

Figure 2: The empirical distribution and our binomial approximation of Hamming distance for two uniformly random vectors on the $n$-hypersphere, conditioned on being separated by an angle $\theta = 15°$. From left to right, $n = 16, 64$. Each empirical distribution was calculated from the results of $10^6$ trials.

### 2.1.2 FORMULATION

With batch size $b$, let $\boldsymbol{Y} = (\boldsymbol{y}^1, \ldots \boldsymbol{y}^b)^T$ be our batch-normalized logit layer for a batch of inputs $(\boldsymbol{x}^1, \ldots \boldsymbol{x}^b)$ and $\boldsymbol{Z} = (\boldsymbol{z}^1, \ldots \boldsymbol{z}^b)^T$ be the $b \times n$ $L^2$-row-normalized version of $\boldsymbol{Y}$; that is, $\boldsymbol{z}^i = \boldsymbol{y}^i/||\boldsymbol{y}^i||_2$. Let $\boldsymbol{P} = \frac{\arccos(\boldsymbol{Z}^T\boldsymbol{Z})}{\pi}$. Let $\boldsymbol{w}$ be the vector of all our model's learnable weights. Let $\boldsymbol{S}$ be a $b \times b$ similarity matrix such that $S_{ij} = 1$ if inputs $\boldsymbol{x}^i$ and $\boldsymbol{x}^j$ are similar and 0 otherwise. Define $\circ$ to be the Hammard product, or pointwise multiplication.

Our loss function is

$$J = -J_1 - \lambda J_2 + \lambda_w J_3$$

with

- $J_1 = \text{Avg}\left[\boldsymbol{S} \circ \log F(r; n, \boldsymbol{P})\right]$, the average log likelihood of each similar pair of inputs to be within Hamming distance $r$.
- $J_2 = \text{Avg}\left[(1 - \boldsymbol{S}) \circ \log F(n - r - 1; n, 1 - \boldsymbol{P})\right]$, the average log likelihood of each dissimilar pair of inputs to be outside Hamming distance $r$.
- $J_3 = ||\boldsymbol{w}||_2^2$, a regularization term on the model's learnable weights to minimize overfitting.

Note that terms $J_1$ and $J_2$, work on all pairwise combinations of images in the batch, providing us with a very accurate estimate of the true gradient.

While most machine learning frameworks do not currently have a binomial CDF operation, many (e.g., Tensorflow and Torch) support a differentiable operation for a beta distribution's CDF. This can be used instead via the well-known relation between the binomial CDF and the beta CDF $I$:

$$F(r; n, p) = I(p; n - r, r + 1)$$

For values of $p$ that are too low, this quantity underflows floating point numbers. This issue can be addressed by a linear extrapolation of log likelihood for $p < p_0$. An exact formula exists, but a simpler approximation suffices, using the fact that $I(p; \alpha, \beta) \propto p^\alpha$ for small $p$:

$$\log(F(r; n, p)) \approx \begin{cases} \log(I(p; n - r, r + 1)), & p \geq p_0 \\ \log(I(p_0; n - r, r + 1)) + \frac{n-r}{p_0}(p - p_0), & p < p_0 \end{cases}$$

### 2.2 TRAINING SCHEME

We construct training batches in a way that ensures every input has another input in the batch it is similar to. Specifically, each batch is composed of groups of $g$ inputs, where each group has one

randomly selected marker input and $g - 1$ random inputs similar to the marker. We then choose $b/g$ random groups to form. During training, similarity between inputs is determined dynamically, such that if two inputs from different groups happen to be similar, they are treated as such.

This method ensures that each loss term is well-defined, since there will be both similar and dissimilar inputs in each batch. Additionally, it provides a better estimate of the true gradient by balancing the huge class of dissimilar inputs with the small class of similar inputs.

### 2.3 Multi-indexing with Embeddings

For additional recall on ANN tasks, we store our model's embedding in each row of the multi-index. We use this to rank results better, returning the closest $l$ of them to the query embedding.This adds to query cost, since evaluating the Euclidean distance between the query's embedding scales with the hash size $n$ and obtaining the top $l$ elements is $O(\log l)$ per result. The heightened query cost allows us to compare query cost against quantization methods, which do the same ranking of final results by embedding distance. When using embeddings to better rank results in this way, we call our method HDT-E.

## 3 Results

### 3.1 ImageNet

We compared HDT against reported numbers for other machine learning approaches to similar image retrieval on ImageNet. We followed the same methodology as Cao et al., using the same training and test sets drawn from 100 ImageNet classes and starting from a pre-trained Inception V3 Szegedy et al. (2015) ImageNet checkpoint accepting $224 \times 224$ images. Fine tuning each model took 5 hours on a single Titan Xp GPU. Following convention, we computed mean average precision (MAP) for the first 1000 results by Hamming distance as our evaluation criterion. We also study our model's precision and recall at different Hamming distances (Figure 3).

We highlight 5 comparator models: DBR-v3 Lu et al. (2017), HashNet Cao et al. (2017), Deep hashing network for efficient similarity retrieval (DHN) Zhu et al. (2016), Iterative Quantization (ITQ) Gong & Lazebnik (2013), and LSH Gionis et al. (99). DBR-v3 learns by first choosing a target hash code for each class to maximize Hamming distance between other target hash codes, then minimizing distance between each image's embedding and target hash code. To the best of our knowledge, it has the highest reported MAP on the ImageNet image retrieval task until this work, partially due to using the Inception V3 architecture whereas previous methods used Alexnet Krizhevsky et al. (2012). HashNet trains a neural network to hash with a supervised cross entropy loss function by gradually sharpening a sigmoid function of its last layer until the outputs are all close to $\pm 1$. DHN similarly trains a neural network with supervised cross entropy loss, but with an added binarization loss term to coerce outputs close to $\pm 1$ instead of sharpening a sigmoid.

We trained a 16-bit model with $r = 2, \lambda = 2000$, a 32-bit model with $r = 2, \lambda = 3000$, and a 64-bit model with $r = 3, \lambda = 3500$. Our method achieved 85.1 to 86.1% MAP (Table 1), a 8.2 to 12.0% absolute improvement over the next best method.

Table 1: ImageNet MAP@1000. Other models' performances are as reported in Lu et al. (2017) and Cao et al. (2017).

| Model | 16 Bits | 32 Bits | 64 Bits |
|---|---|---|---|
| HDT + Inception V3 | **85.3**% | **86.1**% | **85.1**% |
| DBR-v3 | 73.3% | 76.1% | 76.9% |
| HashNet | 50.6% | 63.1% | 68.4% |
| DHN | 31.1% | 47.2% | 57.3% |
| ITQ | 32.3% | 46.2% | 55.2% |
| LSH | 10.1% | 23.5% | 36.0% |

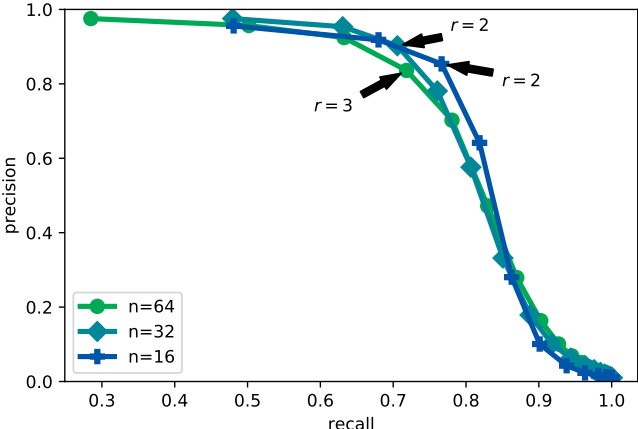

Figure 3: ImageNet precision and recall at different hash lengths for chosen Hamming radii using HDT + Inception V3. Note that at their target Hamming radii, all models achieve similar recall and precision.

Most interestingly, HDT performed better worse on 64-bit hashes than it did on 32-bit hashes. A shorter hash should be strictly worse, since it can be padded with constant bits to a longer hash. Our result may reflect a capacity for the model to overfit slightly with larger bit lengths, an increased difficulty to train a larger model, or a need to better tune parameters. In any case, the clear implication is that 100 ImageNet classes can be encoded in a small number of bits. Even 16-bit binary hashes offer $2^{16}/100 \approx 655$ possibilities per ImageNet class used, generally enough room for each class to own all 137 hashes within a Hamming radius of 2 around its centroid.

## 3.2 SIFT 1M

We compared HDT against the state-of-the-art embedding-to-results method of Product Quantization on the SIFT 1M dataset, which consists of $10^6$ dataset vectors, $10^5$ training vectors, and $10^4$ query vectors in $\mathbb{R}^{128}$.

We trained HDT from scratch using a simple 3-layer Densenet Huang et al. (2017) with 256 relu-activated batch-normalized units per layer. During training, we defined input $x^i$ to be similar to $x^j$ if $x^j$ is among the 10 nearest neighbors to $x^i$. Training each model took 75 minutes on a single Geforce 1080 GPU. We compared the recall-query cost tradeoff at different values of $n$, $r$, and $\lambda$ (Table 2). We used the standard recall metric for this dataset of recall@100, where recall@$k$ is the proportion of queries whose single nearest neighbor is in the top $k$ results.

HDT-E defied even our expectations by providing higher recall than reported numbers for PQ while requiring fewer distance comparisons (Figure 4). This implies that even on embedding-to-result tasks, HDT-E can be implemented to provide better results than PQ with faster query speeds. The improvement is particularly great in the high-recall regime. Notably, HDT-E gets 78.1% recall with an average of 12,709 distance comparisons, whereas PQ gets only 74.4% recall with 101,158 comparisons.

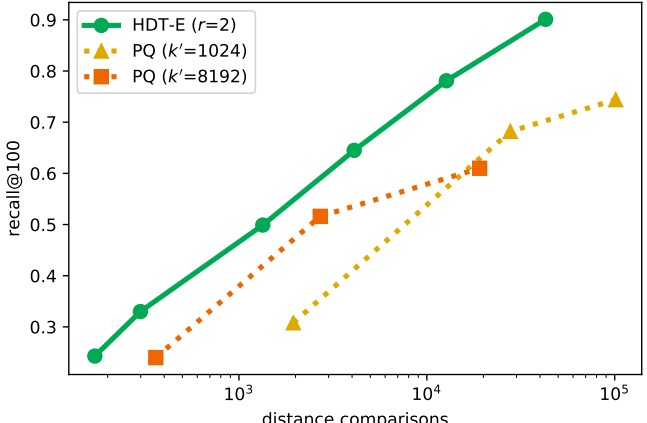

Figure 4: Comparison of HDT-E and PQ 64-bit codes. Metrics used are SIFT 1M recall@100 vs. number of distance comparisons, a measure of query cost. PQ curves are sampled at different parameters for $w \in \{1, 8, 64\}$, the number of centroids whose elements to check against the query. HDT curves are sampled for $\lambda \in \{30000, 10000, 3000, 1000, 300, 100\}$, the loss ratio for false positives.

Table 2: HDT-E SIFT 1M average recall and average number of distance comparisons made with at different values of bits per hash $(n)$, Hamming distance target and Hamming threshold $(r)$, and loss ratio for false positives $(\lambda)$.

| $n$ | $r$ | $\lambda = 100$ | $\lambda = 300$ | $\lambda = 1000$ |
|---|---|---|---|---|
| 16 | 0 | 32.4%, 1463 | 20.6%, 366 | 12.0%, 80.6 |
| 32 | 1 | 59.4%, 4984 | 42.0%, 1324 | 26.5%, 247 |
| 64 | 2 | 90.1%, 42851 | 78.1%, 12709 | 64.5%, 4105 |

## 4 DISCUSSION

Our novel method of Hamming distance targets vastly improved recall and query speed in competitive benchmarks for both data-to-results tasks and embedding-to-results tasks. HDT is also general enough to use any differentiable model and similarity criterion, with applications in image, video, audio, and text retrieval.

We developed a sound statistical model as the foundation of HDT's loss function. We also shed light on why $L^2$-normalization of layer outputs improves learning in conjunction with batch norm. For future study, we are interested in better understanding the theoretical distribution of Hamming distances between points on a sphere separated by a fixed angle.

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
