# OpenReview forum: "Learning Hash Codes via Hamming Distance Targets"
_ICLR.cc/2019/Conference_

### Official Review · AnonReviewer2 · 2018-10-20
**Is it fair to compare the proposed algorithm to existing hashing algorithms and PQ**

**Rating:** 4
**Confidence:** 5

**Review:**

This paper proposed a new hashing algorithm with a new loss function. A multi-indexing scheme is adopted for search.  There is one key issue: in general hashing is not good at multi-indexing search for vector-based search in the Euclidean distance or Cosine similarity. The advantage of hashing is reducing the code size and thus memory cost, but it is still not as good as quantization=based approach.

Here are comments about the experiments.
(1) Table 1: do other algorithms also use multi-indexing or simply linear scan?
(2)  Figure 4: HDT-E is better than PQ. It is not understandable. Something important is missing. How is the search conducted for PQ? Is multi-indexing used? It is also strange to compare the recall in terms of #(distance comparisons).

---

> ### Author Response · Authors · 2018-11-06
> **Yes**
>
> "There is one key issue: in general hashing is not good at multi-indexing search for vector-based search in the Euclidean distance or Cosine similarity.": All multi-index search relies on hash codes, so we are not quite sure what you mean. You may argue that multi-indexing has underperformed on Euclidean or Cosine similarity tasks in the past, but it should be clear from our abstract that our approach (HDT) refutes that.
>
> "The advantage of hashing is reducing the code size and thus memory cost, but it is still not as good as quantization=based approach." - Quantization approaches also create hash codes; to quote Jegou et. al.'s Product Quantization paper, "Formally, a quantizer is a function q mapping a D- dimensional vector x ∈ RD to a vector q(x) ∈ C = {ci;i ∈ I}, where the index set I is from now on assumed to be finite: I = 0...k − 1." Again, as stated in our abstract abstract, our method outperforms Product Quantization on its own benchmark.
>
> (1). The MAP@1000 criterion is defined based on the top 1000 results by Hamming distance (section 3.1). This is achieved for all models by a linear scan.
> (2). The product quantization search is as defined in Jegou et. al.; it does not use multi-indexing. Our HDT-E does use multi-indexing. Recall is not the number of distance comparisons; to quote section 3.2, "Metrics used are SIFT 1M recall@100 vs. number of distance comparisons, a measure of query cost".

---

### Official Review · AnonReviewer1 · 2018-11-02
**Solid idea in the learning to hash area, needs further development**

**Rating:** 6
**Confidence:** 3

**Review:**

Summary: This paper contributes to the area of learning to hash. The goal is to take high-dimensional vectors in R^n resulting from an embedding and map them to binary codewords with the goal of similar vectors being mapped to close codewords (in Hamming distance). The authors introduce a loss function for this problem that's based on angles between points on the hypersphere, relying on the intuition that angles corresponds to the number of times needed to cross to the other side of the hypersphere in each coordinate. This is approximately the Hamming distance under a simple quantization scheme. The loss function itself forces similar points together and dissimilar points apart by matching the Hamming distance to the binomial CDF of the angle quantization. They also suggest a batching scheme that enforces the presence of both similar and dissimilar matches. To confirm the utility of this loss function, the authors empirically verify similarity on ImageNet and SIFT.

Strengths: The main idea, to match up angles between points on the hypersphere and Hamming distance is pretty clever. The loss function itself seems generally useful.

Weaknesses: First, I thought the paper was pretty difficult to understand without a lot of background from previous papers. For the most part the authors don't actually state what the input/output/goals are, leaving it implied from the context, which is tough for the reader. The overall organization isn't great. The paper doesn't contain any theory even for simplified or toy cases (which actually seems potentially tractable here); there is only simple intuition. I think that is fine, but then the empirical results should be extensive, and unfortunately they are not.

Verdict: I think this work contains a great main idea and could become quite a good paper in the future, but the work required to illustrate and demonstrate the idea is not fully there yet.


Comments and Questions:

- Why do you actually need the embedded points y to be on the unit hypersphere? You could compute distances between points at different radii. The results probably shouldn't change much.

- There's at least a few other papers that use a similar idea, for example
Gong et al "Angular Quantization-based Binary Codes for Fast Similarity Search" at NIPS 2012. Would be good to discuss the differences.

- The experimental section seems very limited for an empirical paper. There's at least a few confusing details, noted below:

- The experimental results for ImageNet comparing against other models are directly taken from those reported by Lu et al. That's fine, but it does mean that it's hard to make comparisons against *any* other paper than the Lu paper. For example, if the selected ImageNet classes are different, then the results of the comparison may well be different. I checked the HashNet paper (Cao et al. 2017), and it papers that their own reported numbers for ImageNet are better than those of the Lu et al paper. That is, I see 0.5059 0.6306 0.6835 for 16/32/64 bit codewords vs Lu's result of 0.442 0.606 0.684, which is quoted in this paper. What's causing this difference? It would probably be a bit less convenient but ultimately better if the results for comparison were reproduced by the authors, and possibly on a different class split compared to the single Lu paper.

- The comparison against PQ should also consider more recent works of the same flavor as PQ, which themselves outperform PQ. For example, "Cartesian k-means" by Norouzi and Fleet, or "Approximate search with quantized sparse representations" by Jain et al. These papers also use the SIFT dataset for their experimental result, so it would be great to compare against them.

---

> ### Author Response · Authors · 2018-11-06
> **Addressing your concerns**
>
> Thank you for your thoughtful review. We can clear up most of your concerns easily:
> * Unfortunately different benchmarks in this field use different optimization criteria, so out of input/output/goals, only output can be clearly defined (and we define it to be a binary hash code). A given problem may choose any specific input, such an image, audio clip, or vector. The goals vary wildly, and our benchmarks used MAP and recall vs. query cost. We will work to make this clearer in the final draft.
> * The datasets we compare against are common choices, and we believe they are sufficient (and our % improvements large enough) for the results to be clear.
>
> - Good question. The advantage of comparing the angle between points (rather than Euclidean distance) is that we are able to build a good statistical approximation of the conditional distribution of Hamming distances. Any such model must assume some distribution for the embedding. We believe a uniform distribution on the hypersphere is most natural, since the binarized hash codes depend only on the direction of the embedding point, not its magnitude.
> - While Gong et al. also work with a hypersphere, their approach is very different, choosing a projection that optimizes binarization loss, rather than log likelihood of falling within a desired Hamming distance, which we argue better represents the true optimization goal. We will mention them in our final version.
> -
> - A public commenter already mentioned this, so please refer to that chain. Cao et al. updated those figures later, after Lu et al. published their paper. We will include the updated figures in our final version. Lu et al.'s results simply compare against Cao et al.'s reported results. Both papers use the same subsets of ImageNet and are directly comparable. We believe using the same dataset splits as previous authors makes our results stronger, showing that we did not cherry-pick the dataset.
> - This is a great suggestion.

---

> > ### Comment · AnonReviewer1 · 2018-11-26
> > **Response to Author's Questions**
> >
> > I appreciate the authors' response. I've increased my score to 6, as most of my smaller questions have been resolved.

---

### Official Review · AnonReviewer3 · 2018-11-06
**Interesting intuition but still far from a real-world solution**

**Rating:** 4
**Confidence:** 3

**Review:**

This paper is about learning to hash. The basic idea is motivated by the intuition: given points z_i and z_j on the hypersphere, the angle between the two points is arccos(z_i \dot z_j), while the probability that a random bit differs between them is arccos(z_i \dot z_j)/\pi. This leads to a nice formulation of learning Hamming Distance Target (HDT), although the optimization procedure requires every input has a similar neighbor in the batch.

The minor issue of this paper is that the writing should be polished. There are numerous typos in paper citing (e.g., Norouzi et al in the 3rd page is missing the reference; Figure 3.2 in the 7th page should be Figure 3; and a number of small typos). But I believe these issues could be fixed easily.

The major issue is how we should evaluate a learning to hash paper with nice intuition but not convincing results. Below are my concerns of the proposed approach.

1. Learning to hash (including the HDT in this paper) and product quantization (PQ) are not based on the same scenario,  so it is unfair to claim hashing method outperforms PQ.

Most learning to hash methods requires two things in the following:
a) the query samples
b) similar/dissimilar samples (or we can call them neighbors and non-neighbor) to the query

PQ does not require a) and b). As a result, in PQ based systems, a query can be compared with codewords using Euclidean distance, without mapping to a hash code. This is important especially for novel queries, because if the system does not see similar samples during training, it will probably fail to map such samples to good hash codes.

Such advantage of PQ (or other related quantization methods) is important for real-world systems, however, not obvious in a controlled experiment setting. As shown in the paper, HDT assumes the queries will be similar in the training and testing stages, and benefits from this restricted setting. But I believe such assumption may not hold in real systems.

2. It is not clear to me that how scalable the proposed method is.

I hope section 1.2 can give analysis on both **space** and time complexity of Algorithm 2. It will be more intuitive to show how many ms it will take to search a billion scale dataset. Currently I am not convinced how scalable the proposed algorithm is.

3. Implementation details
In page 5, it is not clear how the hyper parameters \lamda, \lamda_w and p_0 are selected and how sensitive the performance is. I am also interested in the comparison with [Johnson Dooze Jegou 2017] “Billion-scale similarity search with GPUs”.

4. Missing literature
I think one important recent paper is “Multiscale quantization for fast similarity search” NIP 2017


To summarize, I like the idea of this paper but I feel there are still gap between the current draft and real working system. I wish the submission could be improved in the future.

---

> ### Author Response · Authors · 2018-11-06
> **Addressing your concerns**
>
> Thank you for your thoughtful review. We have used HDT in production systems with datasets of over 10 Billion rows and achieved average retrieval times of <2ms, so it is quite practical. We can clear up most of your concerns easily:
>
> 1. The comparison against PQ is fair because both are provided only the "train" set from the SIFT1M dataset; neither receives the query samples. As described in section 3.2, we defined similarity within the train set by each elements 10 nearest neighbors in the train set, but this is not additional information.
>
> PQ also "learns" a codebook, relying on the assumption that training and testing datasets will be similar. There may be cases where it better handles novel data points, but HDT performed much better on a common benchmark nonetheless.
>
> 2. As mentioned, we have used HDT with datasets of over 10 Billion rows and achieved average retrieval times of <2ms. Expected memory usage for kNN is simply O(k) by streaming through query results. Expected space to store the indexed dataset is O((r+1)Nn), where r is the Hamming radius and n is the number of bits per hash. Since r is practically no more than ~5, and hash bit size is small, this is not much of a concern.
>
> 3. We plot different values of \lambda in Figure 4 to demonstrate how the recall/performance tradeoff varies, so that should address part of your question. p_0 is simply chosen such to extrapolate values of likelihood less that 10^{-100}. As \lambda_w is simply a regularization term, we did think it warranted an entire table or plot to compare.
>
> How would you recommend comparing against “Billion-scale similarity search with GPUs”? We are familiar with this work, but its main feature and results are for exact kNN with powerful computing resources, so there is no interesting recall/performance comparison to draw.
>
> 4. We have just included this in the latest draft.

---

> > ### Comment · AnonReviewer3 · 2018-11-26
> > **thanks**
> >
> > I still feel the proposed method is a supervised algorithm (with similarity/dissimilarity labels) while PQ is unsupervised. Seems the other reviewers have similar opinions so the future version could clarify this.
> >
> > I would like to raise my rating to marginally above acceptance if the paper can describe convincingly how the algorithm is used in products, especially of both precision/recall and speed.

---

> > > ### Author Response · Authors · 2018-11-26
> > > **Supervised clarification and practical precision/recall/speed**
> > >
> > > You are absolutely correct that PQ is typically used as an unsupervised method. In this context of learning to hash, supervised is often used to mean that the model learned from the same dataset that queries are made against, whereas unsupervised is used to mean that the model learned from a different dataset. In our SIFT 1M example, both PQ and HDT-E are trained in an unsupervised way.
> > >
> > > We do indeed use similarity labels, but we make them by discretizing the distances between SIFT vectors into 0's and 1's, which is strictly less information than the exact distances used in training PQ. We assert that discretizing these distances should not be considered applying additional labeled data. We will clarify this in the paper.
> > >
> > > We are currently using HDT on a growing dataset of over 1 million hours of video (>1 petabyte), hashing it at 6 to 16 FPS for over 20 billion hashes with 32-bit hash substrings. Our use case is reverse video lookup. Every hour, we query over 10 million frames of incoming video against this multi-indexed dataset, using a small database instance (6 nodes) on commodity hardware.
> > >
> > > On a per-frame level, we achieved 81% recall and a false positive rate of 10^-10. By considering nearby frames, our recall is somewhat higher, and precision is unmeasurably close to 1.
> > >
> > > Since this is a near-neighbor task, high performance is synonymous with low frame-wise false positive rate. This product has seen substantial iteration. Our original implementation using a wavelet embedding had a per-frame recall of 70% and a false positive rate of 9x10^-8. While we could usually filter out false positives by considering nearby frames, returning hundreds of them slowed performance by a factor of roughly 10. When we switched to a hash trained with HDT, this problem dissolved.

---

### Public Comment · (anonymous) · 2018-09-30
**Some concerns regarding ImageNet-100 results**

Dear authors, I have some concerns for the ImageNet-100 results in your paper:

(1)  Results from other methods such as the popular HashNet are not the same as reported in the original paper (for example 16 bits mAP@1000)

(2) HashNet model use pretrained AlexNet whereas your model use pretrained ResNet 50. While I agree that ResNet is a much better choice, the choice of the model is very important for the final performance. For fair comparison, you should include results from AlexNet.

(3)  Resutls for ImageNet mAP@1000 goes down as number of bits increases, this does not seem right to me.

(4)  What about multi-label datasets such as MS COCO or NUS-Wide?

Thank you.

---

> ### Author Response · Authors · 2018-09-30
> **Addressing your ImageNet-100 concerns**
>
> We tried to be as consistent as possible with the literature, and are currently working on adding more comparison datasets (i.e., MS COCO and NUS-Wide).
>
> (1) As we mentioned, our comparison numbers are drawn from the Lu et. al.'s DBR paper. We will switch to use the better of Cao et. al.'s results and Lu et. al.'s.
>
> (2) We chose ResNet 50 since there were already a few models being used in the literature, like DBR-v3 using InceptionV3 (a larger model with higher ImageNet accuracy than ResNet 50). We understand your concern, though, and will update with AlexNet results as soon as possible.
>
> (3) We address this thoroughly at the end of section 3.1.
>
> (4) Our method works for mutli-label datasets as well, since we can use any similarity matrix. As mentioned, we are working on adding more datasets.
>
> Thank you

---

> > ### Public Comment · (anonymous) · 2018-10-06
> > **Inception V3 will be fine**
> >
> > Thank you for your response. I think you should at least include results trained on Inception V3 since this is the one you are mainly comparing to. AlexNet may not be necessary if results on Inception V3 is even better. The choice of ResNet V2 50 just seems odd. No other baseline papers currently use it. Add why not the original ResNet or ResNet-101? It makes it look like a hand picked model.

---

> > > ### Author Response · Authors · 2018-10-18
> > > **Inception V3 results**
> > >
> > > I have run the Inception V3 benchmarks:
> > >
> > > 16 bits: 85.3% MAP
> > > 32 bits: 86.1% MAP
> > > 64 bits: 85.1% MAP
> > >
> > > As expected, this increased our MAP slightly, making our main result slightly more impressive.
> > > We are still working on adding Alexnet benchmarks.

---

### Meta-Review · Area_Chair1 · 2018-12-14
**The paper needs improvement**

**Confidence:** 4
**Recommendation:** Reject

**Metareview:**

The paper proposes learning a hash function that maps high dimensional data to binary codes, and uses multi-index hashing for efficient retrieval. The paper discusses similar results to "Similarity estimation techniques from rounding algorithms, M Charikar, 2002" without citing this paper. The proposed learning idea is also similar to "Binary Reconstructive Embedding, B. Kulis, T. Darrell, NIPS'09" without citation. Please study the learning to hash literature and discuss the similarities and differences with your approach.

Due to missing citations and lack of novelty, I believe the paper does not pass the bar for acceptance at ICLR.


PS: PQ and its better variants (optimized PQ and cartesian k-means) are from a different family of quantization techniques as pointed out by R3 and multi-index hashing is not directly applicable to such techniques. Regardless, I am also surprised that your technique just using hamming distance is able to outperform PQ using lookup table distance.